# Robust Optimization in Causal Models and $G$-Causal Normalizing Flows

## Abstract

In this paper, we show that interventionally robust optimization problems in causal models are continuous under the $G$-causal Wasserstein distance, but may be discontinuous under the standard Wasserstein distance. This highlights the importance of using generative models that respect the causal structure when augmenting data for such tasks. To this end, we propose a new normalizing flow architecture that satisfies a universal approximation property for causal structural models and can be efficiently trained to minimize the $G$-causal Wasserstein distance. Empirically, we demonstrate that our model outperforms standard (non-causal) generative models in data augmentation for causal regression and mean-variance portfolio optimization in causal factor models.

## 1 Introduction

Solving optimization problems often requires generative data augmentation (Chen et al., 2024; Zheng et al., 2023), particularly when out-of-sample distributional shifts are expected to be frequent and severe, as in the case of financial applications. In such cases, only the most recent data points are representative enough to be used in solving downstream tasks (such as hedging, regression or portfolio selection), resulting in small datasets that require generative data augmentation to avoid overfitting (Bailey et al., 2017). However, when using generative models for data augmentation, it is essential to choose their training loss in a way that is compatible with the downstream tasks, so as to guarantee good and stable performance.

It is well-known, for instance, that multi-stage stochastic optimization problems are continuous under the *adapted* Wasserstein distance, while they may be discontinuous under the standard Wasserstein distance (Pflug & Pichler, 2012; 2014; Backhoff-Veraguas et al., 2020). This insight prompted several authors to propose new time-series generative models that attempt to minimize the adapted Wasserstein distance, either partially (Xu et al., 2020) or its one-sided[1] version (Acciaio et al., 2024).

In this paper we prove a generalization of this result for causal models. Specifically, we show that causal optimization problems (i.e. problems in which the control variables can depend only on the parents of the state variables in the underlying causal DAG $G$) are continuous with respect to the $G$-causal Wasserstein distance (Cheridito & Eckstein, 2025).

Furthermore, we prove that solutions to $G$-causal optimization problems are always interventionally robust. This means that causal optimization can be understood as a way of performing Distributionally Robust Optimization (DRO) (Chen et al., 2020; Kuhn et al., 2025) by taking into account the problem's causal structure.

Next, we address the challenge of designing a generative model capable of good approximations under the $G$-causal Wasserstein distance. We radically depart from existing approaches for the adapted Wasserstein distance and propose a novel $G$-causal normalizing flow model based on invertible neural couplings that respect the causal structure of the data. We prove a universal approximation property for this model class and that maximum likelihood training indeed leads to distributions that are

---

[1] Also known in the literature as the causal Wasserstein distance, because it respects the temporal flow of information in the causal direction (from past to present). This terminology conflicts with the way the term "causal" is used in causal modelling. To avoid misunderstandings we talk of the "$G$-causal" Wasserstein distance and refer to the causal Wasserstein distance as the "one-sided" adapted Wasserstein distance.

close to the target distribution in the $G$-causal Wasserstein distance. Since the standard, adapted and CO-OT Wasserstein distances are all special cases of the $G$-causal Wasserstein distance, this model family provides optimal generative augmentation models for a vast class of empirical applications.

**Contributions.** Our main contributions are the following:

- We prove that causal optimization problems (i.e. problems in which optimizers must be functions of the state variables' parents in the causal DAG $G$) are continuous under the $G$-causal Wasserstein distance, but may be discontinuous under the standard Wasserstein distance.

- We prove that solutions to $G$-causal optimization problems are always interventionally robust.

- We introduce $G$-causal normalizing flows and we prove that they satisfy a universal approximation property for causal structural models under very mild conditions.

- We prove that $G$-causal normalizing flows minimize the $G$-causal Wasserstein distance between data and model distribution by simple likelihood maximization.

- We show empirically that $G$-causal normalizing flows outperform non-causal generative models (such as variational auto-encoders, standard normalizing flows, and nearest-neighbor KDE) when used to perform generative data augmentation in two empirical setups: causal regression and mean-variance portfolio optimization in causal factor models.

## 2 BACKGROUND

**Notation.** We denote by $\|\cdot\|$ the Euclidean norm on $\mathbb{R}^d$ and by $L^p(\mu)$ the space $L^p(\mathbb{R}^d, \mathcal{B}(\mathbb{R}^d), \mu)$ equipped with the norm $\|f\|_{L^p(\mu)} := \left(\int_{\mathbb{R}^d} \|f(z)\|^p \mu(dz)\right)^{1/p}$. $\mathcal{P}(\mathbb{R}^d)$ denotes the space of all Borel probability measures on $\mathbb{R}^d$. $\mathcal{N}(\mu, \Sigma)$ is the multivariate Gaussian distribution with mean $\mu$ and covariance matrix $\Sigma$, $\mathcal{U}([0,1]^d)$ is the uniform distribution on the $d$-dimensional hypercube, $I_d$ denotes the $d \times d$ identity matrix.

We use set-indices to slice vectors, i.e. if $x = (x_1, \ldots, x_d) \in \mathbb{R}^d$ and $A \subseteq \{1, \ldots, d\}$, then $x_A := (x_i, i \in A) \in \mathbb{R}^{|A|}$. If $\mu \in \mathcal{P}(\mathbb{R}^d)$ and $X = (X_1, \ldots, X_d) \sim \mu$, then the regular conditional distribution of $X_A$ given $X_B$ is denoted by $\mu(dx_A | x_B)$, for all $A, B \subseteq \{1, \ldots, d\}$ with $A \cap B = \emptyset$.

### 2.1 STRUCTURAL CAUSAL MODELS

We assume throughout that $G = (V, E)$ is a given directed acyclic graph (DAG) with a finite index set $V = \{1, \ldots, d\}$, which we assume, without loss of generality, to be sorted (i.e. $(i, j) \in E$, then $i < j$). If $A \subseteq V$, we denote by $\text{PA}(A) := \{i \in V \setminus A \mid \exists j \in A \mid (i, j) \in E\}$ the set of parents of the vertices in $A$ (notice that $\text{PA}(A) \subseteq V \setminus A$ by definition).

In this paper, we work with structural causal models, as presented in Peters et al. (2017).

**Definition 2.1** (Structural Causal Model (SCM)). Given a DAG $G = (V, E)$, a Structural Causal Model (SCM) is a collection of assignments

$$X_i := f_i(X_{\text{PA}(i)}, U_i), \quad \text{for all } i = 1, \ldots, d,$$

where the noise variables $(U_i, i = 1, \ldots, d)$ are mutually independent.

### 2.2 $G$-CAUSAL WASSERSTEIN DISTANCE

**Definition 2.2** (G-compatible distribution). A distribution $\mu \in \mathcal{P}(\mathbb{R}^d)$ is said to be $G$-compatible, and we denote it by $\mu \in \mathcal{P}_G(\mathbb{R}^d)$, if any of the following equivalent conditions holds:

1. there exist a random vector $X = (X_1, \ldots, X_d) \sim \mu$ together with measurable functions $f_i : \mathbb{R}^{|\text{PA}(i)|} \times \mathbb{R} \to \mathbb{R}, (i = 1, \ldots, n)$, and mutually independent random variables $(U_i, i = 1, \ldots, d)$ such that

$$X_i = f_i(X_{\text{PA}(i)}, U_i), \quad \text{for all } i = 1, \ldots, d.$$

2. For every $X \sim \mu$, one has

$$X_i \perp\!\!\!\perp X_{1:i-1} \mid X_{\text{PA}(i)}, \quad \text{for all } i = 2, \ldots, d.$$

3. The distribution $\mu$ admits the following disintegration:

$$\mu(dx_1, \ldots, dx_d) = \prod_{i=1}^{d} \mu(dx_i \mid x_{\text{PA}(i)}).$$

For a proof of the equivalence of these three conditions, see Cheridito & Eckstein (2025, Remark 3.2).

**Definition 2.3** ($G$-bicausal couplings). A coupling $\pi \in \Pi(\mu, \nu)$ between two distributions $\mu, \nu \in \mathcal{P}_G(\mathbb{R}^d)$ is $G$-**causal** if there exist $(X, X') \sim \pi$ such that

$$X'_i = g_i(X_i, X_{\text{PA}(i)}, X'_{\text{PA}(i)}, U_i)$$

for some measurable mappings $(g_i)_{i=1}^d$ and mutually independent random variables $(U_i)_{i=1}^d$. If also the distribution of $(X', X)$ is $G$-causal, then we say that $\pi$ is $G$-**bicausal**. We denote by $\Pi_G^{\text{bc}}(\mu, \nu)$ the set of all $G$-bicausal couplings between $\mu$ and $\nu$.

**Definition 2.4** ($G$-causal Wasserstein distance). Denote by $\mathcal{P}_{G,1}(\mathbb{R}^d)$ the space of all $G$-compatible distributions with finite first moments. Then the $G$-causal Wasserstein distance between $\mu, \nu \in \mathcal{P}_{G,1}(\mathbb{R}^d)$ is defined as:

$$W_G(\mu, \nu) := \inf_{\pi \in \Pi_G^{\text{bc}}(\mu, \nu)} \int_{\mathbb{R}^d \times \mathbb{R}^d} \|x - x'\| \, \pi(dx, dx').$$

Furthermore, $W_G$ defines a semi-metric on the space $\mathcal{P}_{G,1}(\mathbb{R}^d)$ (Cheridito & Eckstein, 2025, Proposition 4.3).

## 3 ROBUST OPTIMIZATION IN STRUCTURAL CAUSAL MODELS

Suppose we are given an SCM $X \sim \mu \in \mathcal{P}_G(\mathbb{R}^d)$ on a DAG $G = (V, E)$ and we want to solve a stochastic optimization problem in which the state variables $X_T$ are specified by a vertex subset $T \subseteq V$ (called the *target set*) and the control variables can potentially be all remaining vertices in the graph, i.e. $X_{V \setminus T}$. To avoid feedback loops between state and control variables, we will need the following technical assumption.

**Assumption 3.1.** The DAG $G = (V, E)$ and the target set $T \subseteq V$ are such that $G$ quotiened by the partition $\{T\} \cup \{\{i\}, i \in V \setminus T\}$ is a DAG.

*Remark* 3.2. Assumption 3.1 is quite mild and is equivalent to asking that if $i, j \in T$, then $X_i$ cannot be the parent of a parent of $X_j$. This guarantees that $\text{PA}(T) \cap \text{CH}(T) = \emptyset$, which is nothing but asking that $X_T$ be part of a valid SCM *as a random vector*, see Fig. 1 and 2.

**Definition 3.3** ($G$-causal function). Given a target set $T \subseteq V$, we say that a function $h : \mathbb{R}^{|V \setminus T|} \to \mathbb{R}^{|T|}$ is $G$-causal (with respect to $T$) if $h$ depends only on the parents of $X_T$, i.e. $h(x) = h(x_{\text{PA}(T)})$, for all $x \in \mathbb{R}^{|V \setminus T|}$.

**Definition 3.4** ($G$-causal optimization problem). Let $G = (V, E)$ be a sorted DAG, $X \sim \mu \in \mathcal{P}_G(\mathbb{R}^d)$ and let $T \subseteq V$ be a target set. If $Q : \mathbb{R}^{|T|} \times \mathbb{R}^{|V \setminus T|} \to \overline{\mathbb{R}}$ is a function to be optimized, then a $G$-causal optimization problem (with respect to $T$) is an optimization problem of the following form:

$$\min_{\substack{h : \mathbb{R}^{|V \setminus T|} \to \mathbb{R}^{|T|} \\ h \text{ is } G\text{-causal}}} \mathbb{E}^\mu \left[ Q(X_T, h(X_{V \setminus T})) \right]. \tag{1}$$

Any minimizer of (1) is called a $G$-causal optimizer.

The following result shows that $G$-causal optimizers are always interventionally robust. This underscores the desirability of $G$-causal optimizers when we expect the data distribution to undergo distributional shifts due to interventions between training and testing time.

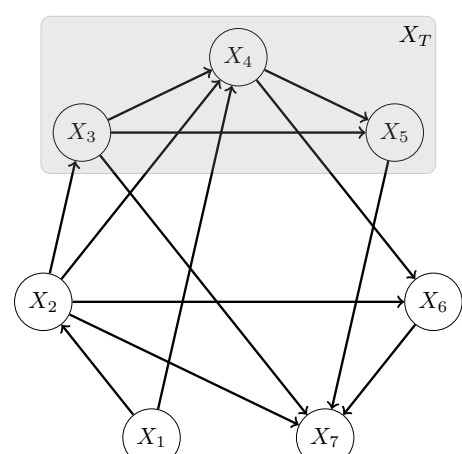 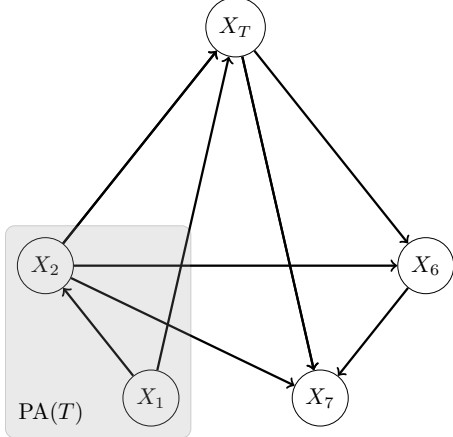

Figure 1: DAG $G$ before quotienting (target set $T$ highlighted).

Figure 2: DAG $G$ after quotienting (vertex set $PA(T)$ highlighted).

**Theorem 3.5** (Robustness of $G$-causal optimizers)**.** *Let $h^*$ be a solution of the problem in Eq. (1). Then:*

$$h^* \in \underset{h:\mathbb{R}^{|V \setminus T|} \to \mathbb{R}^{|T|}}{\arg\min} \sup_{\nu \in \mathcal{I}(\mu)} \mathbb{E}^{\nu} \left[ Q(X_T, h(X_{V \setminus T})) \right],$$

*where*

$$\mathcal{I}(\mu) := \{\nu \in \mathcal{P}(\mathbb{R}^d) \mid \nu(dx_T | x_{PA(T)}) = \mu(dx_T | x_{PA(T)}) \text{ and } \mathrm{supp}(\nu(dx_{PA(T)})) \subseteq \mathrm{supp}(\mu(dx_{PA(T)}))\}$$

*is the set of all interventional distributions that leave the causal mechanism of $X_T$ unchanged.*

*Proof.* It's enough to show that for any $h : \mathbb{R}^{|V \setminus T|} \to \mathbb{R}^{|T|}$ and any $\nu \in \mathcal{I}(\mu)$, there exists a $\nu' \in \mathcal{I}(\mu)$ such that $\mathbb{E}^{\nu'} \left[ Q(X_T, h(X_{V \setminus T})) \right] \geq \mathbb{E}^{\nu} \left[ Q(X_T, h^*(X_{PA(T)})) \right]$.

Given $\nu \in \mathcal{I}(\mu)$, define $\nu'(dx) := \nu(dx_{V \setminus (T \cup PA(T))})\nu(dx_{PA(T)}, dx_T)$. Then:

$$\mathbb{E}^{\nu'} \left[ Q(X_T, h(X_{V \setminus T})) \right] = \int \nu(dx_{V \setminus (T \cup PA(T))}) \int \nu(dx_{PA(T)}, dx_T)Q(x_T, h(x_{V \setminus T}))$$

$$= \int \nu(dx_{V \setminus (T \cup PA(T))}) \int \nu(x_{PA(T)}) \int \mu(dx_T \mid x_{PA(T)})Q(x_T, h(x_{V \setminus T}))$$

$$\geq \int \nu(dx_{V \setminus (T \cup PA(T))}) \int \nu(x_{PA(T)}) \int \mu(dx_T \mid x_{PA(T)})Q(x_T, h^*(x_{PA(T)}))$$

$$= \mathbb{E}^{\nu} \left[ Q(X_T, h^*(X_{PA(T)})) \right]$$

where the second equality follows from $\nu \in \mathcal{I}(\mu)$ and the inequality follows from Eq. (1), Lemma A.1, and $\mathrm{supp}(\nu(dx_{PA(Y)})) \subseteq \mathrm{supp}(\mu(dx_{PA(Y)}))$. $\qquad\square$

*Remark* 3.6. The theorem above is a generalization of (Rojas-Carulla et al., 2018, Theorem 4), which covered the mean squared loss only. We explicitly added the assumption $\mathrm{supp}(\nu(dx_{PA(Y)})) \subseteq \mathrm{supp}(\mu(dx_{PA(Y)}))$, for all $\nu \in \mathcal{I}(\mu)$, which is needed also for their theorem to hold.

The next theorem shows that the value functionals of $G$-causal optimization problems are continuous with respect to the $G$-causal Wasserstein distance, while they may fail to be continuous with respect to the standard Wasserstein distance (as we show in Example 3.8 below). This proves that the $G$-causal Wasserstein distance is the right distance to control errors in causal optimization problems and, in particular, interventionally robust optimization problems.

**Theorem 3.7** (Continuity of $G$-causal optimization problems)**.** *Let $G = (V, E)$ be a sorted DAG, $X \sim \mu \in \mathcal{P}_G(\mathbb{R}^d)$ and let $T \subseteq V$ be a target set, such that Assumption 3.1 holds. If $Q : \mathbb{R}^{|T|} \times$*

$\mathbb{R}^{|V \setminus T|} \to \overline{\mathbb{R}}$ *is such that $x \mapsto Q(x, h)$ is locally L-Lipschitz (uniformly in h) and $h \mapsto Q(x, h)$ is convex, then the value functional*

$$\mu \mapsto \mathcal{V}(\mu) := \min_{\substack{h:\mathbb{R}^{|V \setminus T|} \to \mathbb{R}^{|T|} \\ h \text{ is } G\text{-causal}}} \mathbb{E}^\mu \left[ Q(X_T, h(X_{V \setminus T})) \right]$$

*is continuous with respect to the G-causal Wasserstein distance.*

*Proof.* See proof in Appendix B.1. □

**Example 3.8.** Define $\mu_\varepsilon \in \mathcal{P}_G(\mathbb{R}^2)$ as the following SCM:

$$\begin{cases} Y := \text{sgn}(X), \\ X := \varepsilon \cdot U, \end{cases} \quad \text{where } U \sim \text{Ra}(1/2),$$

where $\text{Ra}(p)$ denoted the Rademacher distribution $p\delta_1 + (1 - p)\delta_{-1}$, and consider the following $G$-causal regression problem:

$$\mathcal{V}(\mu) = \inf_{\substack{h:\mathbb{R} \to \mathbb{R} \\ h \ G\text{-causal}}} \mathbb{E}^\mu \left[ (Y - h(X))^2 \right].$$

Then as $\varepsilon \to 0$ we have that $\mu_\varepsilon = \frac{1}{2}\delta_{(\varepsilon,1)} + \frac{1}{2}\delta_{(-\varepsilon,-1)}$ converges to $\mu := \frac{1}{2}\delta_{(0,1)} + \frac{1}{2}\delta_{(0,-1)} = \delta_0 \otimes \text{Ra}(1/2)$ under the standard Wasserstein distance, but $\lim_{\varepsilon \to 0} \mathcal{V}(\mu_\varepsilon) = 0 \neq 1 = \mathcal{V}(\mu)$.

# 4 PROPOSED METHOD: $G$-CAUSAL NORMALIZING FLOWS

Theorem 3.7 and Example 3.8 imply that generative augmentation models that are not trained under the $G$-causal Wasserstein distance may lead to optimizers that severely underperform on $G$-causal downstream tasks. To solve this issue, we propose a novel normalizing flow architecture capable of minimizing the $G$-causal Wasserstein distance from any data distribution $\mu \in \mathcal{P}_G(\mathbb{R}^d)$. Since the standard, adapted and CO-OT Wasserstein distances are all special cases of the $G$-causal Wasserstein distance, this model family provides optimal generative augmentation models for a vast class of empirical applications.

A $G$-causal normalizing flow $\hat{T} = \hat{T}^{(d)} \circ \cdots \circ \hat{T}^{(1)}$ is a composition of $d$ neural coupling flows $\hat{T}^{(k)} : \mathbb{R}^d \to \mathbb{R}^d$ of the following form:

$$\hat{T}_i^{(k)}(x) = \begin{cases} g(x_i; \theta(x_{\text{PA}(i)})) & \text{if } i = k \\ \text{id} & \text{if } i \neq k \end{cases} \quad (2)$$

where $g : \mathbb{R} \times \Theta(n) \to \mathbb{R}$ is a shallow MLP of the form:

$$g(x, \theta) = \sum_{i=1}^n w_i^{(2)} \rho(w_i^{(1)} x + b_i^{(1)}) + b^{(2)} \quad (3)$$

with parameters $\theta := (w^{(1)}, b^{(1)}, w^{(2)}, b^{(2)}) \in \Theta(n) := \mathbb{R}_{>0}^n \times \mathbb{R}^n \times \mathbb{R}_{>0}^n \times \mathbb{R}$ and custom activation function[2]:

$$\rho(x) = \frac{1}{2}\text{LeakyReLU}_{\alpha^{-1}}(1 + x) - \frac{1}{2}\text{LeakyReLU}_{\alpha^{-1}}(1 - x), \quad \alpha \in (0, 1). \quad (4)$$

We denote by $\text{IncrMLP}(n)$ the class of all MLPs with $n$ hidden neurons and parameter space $\Theta(n)$. It is easy to see that $\text{IncrMLP}(n)$ contains only continuous, piecewise linear, strictly increasing (and, therefore, *invertible*) functions, thanks to the choice of activation function[3] and parameter space. The inverse of $g$ and its derivative can be computed efficiently, which allows the coupling flow in Eq. (2) to be easily implemented in a normalizing flow model (see code in the supplementary material).

---

[2]Recall that the LeakyReLU activation function is defined as $\text{LeakyReLU}_\alpha(x) := x\mathbb{1}_{\{x \geq 0\}} + \alpha x\mathbb{1}_{\{x < 0\}}$.

[3]One cannot just take $\rho(x) = \text{ReLU}(x)$, because $g$ could fail to be strictly increasing, nor $\rho(x) = \text{LeakyReLU}_\alpha(x)$, because then $g$ would be constrained to be convex, which harms model capacity.

In Eq. (2) we specify the parameters of $g$ in terms of a function $\theta(x_{\text{PA}(i)})$, which we take to be an MLP[4]. The particular choice of MLP class does not matter, as long as the assumptions of (Leshno et al., 1993, Theorem 1) are satisfied[5] and we denote by MLP any such class. Since the outputs of $\theta(\cdot) \in$ MLP are used as parameters for another MLP, $g(\cdot)$, it is common to say that $\theta(\cdot)$ is a *hypernetwork* (Chauhan et al., 2024). Therefore we say that the coupling flow in Eq. (2) is a hypercoupling flow and we denote by HyperCpl$(n, \theta(\cdot))$ the class of hypercoupling flows with $g(\cdot) \in$ IncrMLP$(n)$ and parameter hypernetwork $\theta(\cdot) \in$ MLP.

Since each hypercoupling flow in a $G$-causal normalizing flow acts only on a subset of the input coordinates it effectively functions as a scale in a multi-scale architecture, thus reducing the computational burden by exploiting our a priori knowledge of the causal DAG $G$.

*Remark* 4.1. We emphasize that the DAG $G$ is an *input* of our model, not an output. We assume, therefore, that the modeler has estimated the causal skeleton $G$, using any of the available methods for causal discovery Nogueira et al. (2022); Zanga et al. (2022). On the other hand, we do not require any knowledge of the functional form of the causal mechanisms, which our model will learn directly from data.

Next, we turn to the task of proving that $G$-causal normalizing flows are universal approximators for structural causal models.

**Definition 4.2** ($G$-compatible transformation.). Let $G$ be a sorted DAG. A map $T : \mathbb{R}^d \to \mathbb{R}^d$ is a $G$-compatible transformation if each coordinate $T_i(x)$ is a function of $(x_i, x_{\text{PA}(i)})$, for all $i = 1, \ldots, d$. Furthermore, a $G$-compatible transformation $T$ is called (strictly) increasing if each coordinate $T_i$ is (strictly) increasing in $x_i$.

**Theorem 4.3.** *Let $\mu \in \mathcal{P}_G(\mathbb{R}^d)$ be an absolutely continuous distribution. Then there exists a $G$-compatible, strictly increasing transformation $T : \mathbb{R}^d \to \mathbb{R}^d$, such that $T_\#\mathcal{U}([0,1]^d) = \mu$.*

*Furthermore, $T$ is of the form $T := T^{(d)} \circ \cdots \circ T^{(1)}$, where each $T^{(k)} : \mathbb{R}^d \to \mathbb{R}^d$ is defined as:*

$$T_i^{(k)}(x) = \begin{cases} F_i^{-1}(x_i \mid x_{PA(i)}) & i = k, \\ id & i \neq k. \end{cases} \quad (k = 1, \ldots, d) \tag{5}$$

*where $F_i^{-1}$ is the (conditional) quantile function of the random variable $X_i \sim \mu(dx_i)$ given its parents $X_{PA(i)} \sim \mu(dx_{PA(i)})$.*

*Proof.* It is easy to check that $T$, as defined, is indeed a $G$-compatible, increasing transformation. The absolute continuity of $\mu$ implies that all conditional distributions admit a density (Jacod & Protter, 2004, Theorem 12.2), therefore a continuous cdf and a strictly monotone quantile function (McNeil et al., 2015, Proposition A.3 (ii)).

Next, we show that $T_\#\mathcal{U}([0,1]^d) = \mu$. By Definition 2.2 we know that there exists $X \sim \mu$ and measurable functions $f_i$ such that $X_i = f_i(X_{\text{PA}}(i), U_i)$ where $U = (U_1, \ldots, U_d)$ is a random vector of mutually independent random variables. Without loss of generality, we can take $U \sim \mathcal{U}([0,1]^d)$ and set $X_i = F_i^{-1}(U_i | X_{\text{PA}}(i))$ (McNeil et al., 2015, Proposition A.6)). $\square$

**Theorem 4.4** (Universal Approximation Property (UAP) for $G$-causal normalizing flows). *Let $\mu \in \mathcal{P}_{G,1}(\mathbb{R}^d)$ be an absolutely continuous distribution with compact support and assume that the conditional cdfs $(x_k, x_{PA(k)}) \mapsto F_k(x_k \mid x_{PA(k)})$ belong to $C^1(\mathbb{R} \times \mathbb{R}^{|PA(k)|})$, for all $k = 1, \ldots, d$.*

*Then $G$-causal normalizing flows with base distribution $\mathcal{U}([0,1]^d)$ are dense in the semi-metric space $(\mathcal{P}_{G,1}(\mathbb{R}^d), W_G)$, i.e. for every $\varepsilon > 0$, there exists a $G$-causal normalizing flow $\hat{T}$ such that*

$$W_G(\mu, \hat{T}_\#\mathcal{U}([0,1]^d)) \leq \varepsilon.$$

*Proof.* See proof in Appendix B.2. $\square$

---

[4]In practice, we enforce $\theta(x_{\text{PA}(i)}) \in \Theta(n)$ by constraining its outputs corresponding to the weights $w^{(1)}$ and $w^{(2)}$ to be strictly positive, either by using a ReLU activation function or by taking their absolute value.

[5]The activation function must be non-polynomial and locally essentially bounded on $\mathbb{R}$. All commonly used activation functions (including ReLU) satisfy this.

*Remark* 4.5. The theorem holds for base distributions other than $\mathcal{U}([0,1]^d)$. In fact any absolutely continuous distribution on $\mathbb{R}^d$ with mutually independent coordinates (such as the standard multivariate Gaussian $\mathcal{N}(0, I_d)$) would work, provided we add a non-trainable layer between the base distribution and the first flow that maps $\mathbb{R}^d$ into the base distribution's quantiles (for $\mathcal{N}(0, I_d)$, such a map is just $\Phi^{\otimes d}$, where $\Phi$ is the standard Gaussian cdf).

In practice $G$-causal normalizing flows are trained using likelihood maximization (or, equivalently, KL minimization), so it is important to make sure that minimizing this loss guarantees that the $G$-causal Wasserstein distance between data and model distribution is also minimized. The following result proves exactly this and is a generalization of Acciaio et al. (2024, Lemma 2.3) and Eckstein & Pammer (2024, Lemma 3.5), which established an analogous claim for the adapted Wasserstein distance.

**Theorem 4.6** ($W_G$ *training via KL minimization*). *Let* $\mu, \nu \in \mathcal{P}_G(K)$ *for some compact* $K \subseteq \mathbb{R}^d$. *Then:*

$$W_G(\mu, \nu) \leq C\sqrt{\frac{1}{2}\mathcal{D}_{KL}(\mu \mid \nu)},$$

*for a constant* $C > 0$.

*Proof.* See proof in Appendix B.3. $\square$

## 5 NUMERICAL EXPERIMENTS

### 5.1 CAUSAL REGRESSION

We study a multivariate causal regression problem of the form:

$$\min_{\substack{h:\mathbb{R}^{|V \setminus T|} \to \mathbb{R}^{|T|} \\ h \text{ is } G\text{-causal}}} \mathbb{E}^{\mu}\left[(X_T - h(X_{V \setminus T}))^2\right], \tag{6}$$

where $\mu \in \mathcal{P}_G(\mathbb{R}^d)$ is a randomly generated linear Gaussian SCM (Peters et al., 2017, Section 7.1.3) with coefficients uniformly sampled in $(-1, 1)$ and homoscedastic noise with unit variance. The sorted DAG $G$ is obtained by randomly sampling an Erdos-Renyi graph on $d$ vertices with edge probability $p$ and eliminating all edges $(i, j)$ with $i > j$.

According to Theorem 3.5, any solution to problem (6) is interventionally robust. In order to showcase this robustness property of the $G$-causal regressor, we compare its performance with that of a standard (i.e. non-causal) regressor when tested out-of-sample on a large number of random soft[6] interventions. Each intervention is obtained by randomly sampling a node $i \in V \setminus T$ and substituting its causal mechanism, $f(X_{\text{PA}(i)}, U_i)$, with a new one, $\tilde{f}(X_{|\text{PA}(i)}, U_i)$. We consider only linear interventions and quantify their interventional strength by computing the following $L^1$-norm:

$$\int \int |f(x_{\text{PA}(i)}, u) - \tilde{f}(x_{\text{PA}(i)}, u)|\mu(dx_{\text{PA}(i)})\lambda(du),$$

where $\mu$ is the original distribution (before intervention) and $\lambda$ is the noise distribution. Interventional strength, therefore, quantifies the out-of-sample variation of the regressor's inputs under the intervention.

We implement a multivariate regression with $d = 10$, $p = 0.5$ and $T = \{5, 6\}$. We report in Fig. 7 and Fig. 8 the worst-case performance of a $G$-causal regressor and of a non-causal regressor (in terms of MSE and $R^2$, respectively) as a function of the interventional strength. At small interventional strengths the non-causal regressor benefits from the information contained in non-parent nodes (which are not available as inputs to the $G$-causal optimizer). These non-parent nodes may belong to the Markov blanket of the target nodes in $G$ and therefore be statistically informative, but their usefulness crucially depends on the stability of their causal mechanisms. As the interventional strength is increased the worst-case performance of the non-causal regressor rapidly deteriorates, while that of the $G$-causal regressor remains stable, as shown in the figures.

---

[6]A soft intervention at a node $i \in V$ leaves its parents and noise distribution unaltered, but changes the functional form of its causal mechanism.

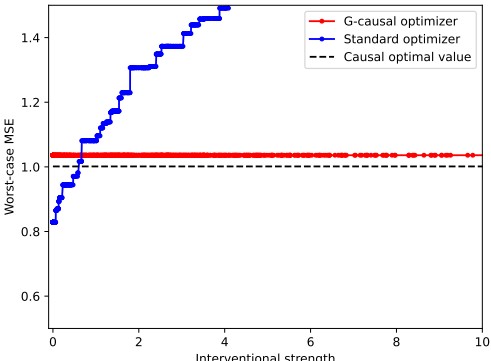

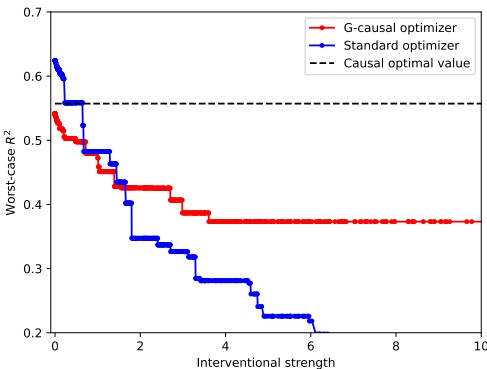

Figure 3: Worst-case MSE vs interventional strength.

Figure 4: Worst-case $R^2$ vs interventional strength.

In Fig. 5 and Fig. 6 we deepen the comparison by plotting the distribution of the performance metrics (MSE and $R^2$, respectively) for both estimators. Notice how interventions deteriorate the performance of the non-causal regressor starting from the least favorable quantiles, while the entire distribution of the performance metrics of the $G$-causal remains stable. These figures also show that the median performance of the causal regressor is, after all, not strongly affected by the linear random interventions we consider. In this sense, non-causal optimizers can still be approximately optimal in applications where distributional shifts are expected to be mild.

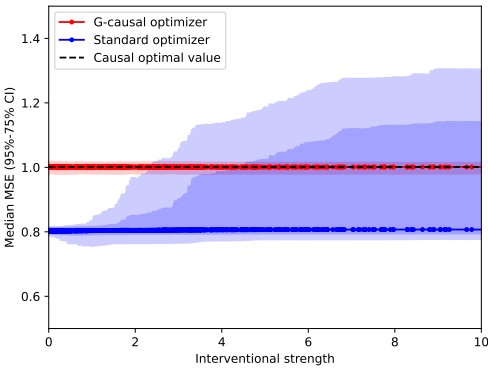

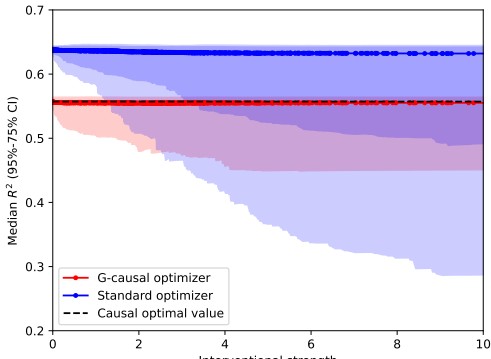

Figure 5: Median and (75%-95%) CI of MSE vs interventional strength.

Figure 6: Median and (75%-95%) CI of $R^2$ vs interventional strength.

Finally, we investigate the performance of our $G$-causal normalizing flow model when used for generative data augmentation. We therefore train several augmentation models (both non-causal and $G$-causal) on a training set of $n = 10000$ samples from $\mu$. We then use them to generate of synthetic training set of $n = 10000$ samples and we train a causal optimizer on it.

As shown in Fig. 7 and Fig. 8, causal optimizers trained using non-causal augmentation models (e.g. RealNVP and VAE) are indeed robust under interventions, but their worst-case metrics are significantly worse than when causal augmentation is used. This is an empirical validation of the fact that the loss used for training the augmentation model plays a crucial role in downstream performance.

## 5.2 CONDITIONAL MEAN-VARIANCE PORTFOLIO OPTIMIZATION

We look at the following conditional mean-variance portfolio optimization problem:

$$\mathcal{V}(\mu) = \inf_{\substack{h:\mathbb{R}^{|V\setminus T|}\to\mathbb{R}^{|T|} \\ h \text{ is } G\text{-causal}}} \left\{ -\mathbb{E}^\mu\left[\langle X_T, h(X_{V\setminus T})\rangle\right] + \frac{\gamma}{2}\text{Var}^\mu\left(\langle X_T, h(X_{V\setminus T})\rangle\right)\right\},$$

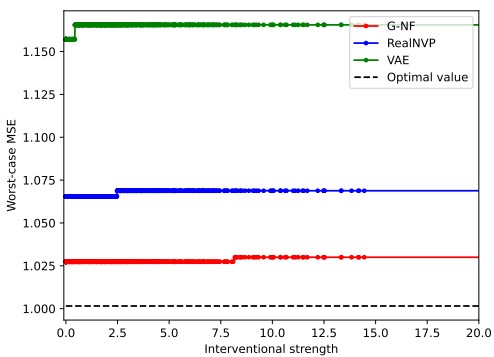

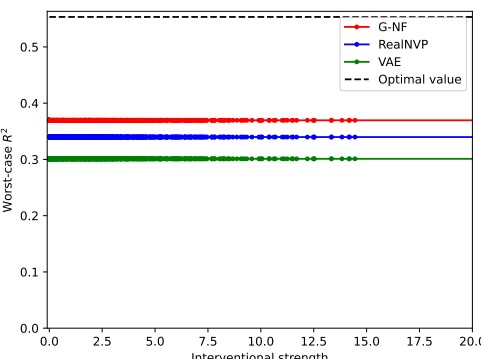

Figure 7: Worst-case MSE after generative data augmentation vs interventional strength.

Figure 8: Worst-case $R^2$ after generative data augmentation vs interventional strength.

where $X \sim \mu \in \mathcal{P}_G(\mathbb{R}^d)$ is a linear Gaussian SCM, with bipartite DAG $G$ with partition $\{T, V \setminus T\}$ and random uniform coefficients in $(-1, 1)$, and $\gamma$ is a given risk aversion parameter. The target variables $X_T$ represent stock returns, while $X_{V \setminus T}$ are market factors or trading signals. We present the results for a high-dimensional example with $|T| = 100$ stocks and $|V \setminus T| = 20$ factors.

We sample random linear interventions exactly as done in the case of causal regression and study empirically the robustness of the $G$-causal portfolio in terms of its Sharpe ratio as the interventional strength increases.

Fig. 9 and Fig. 10 show that the Sharpe ratio of the $G$-causal portfolio is indeed robust to a wide range of interventions, while the performance of non-causal portfolios deteriorates rapidly, starting from the least favorable quantiles.

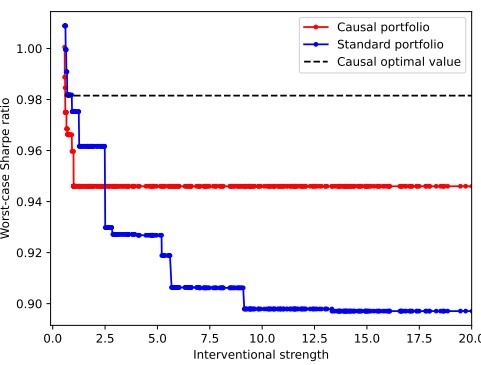

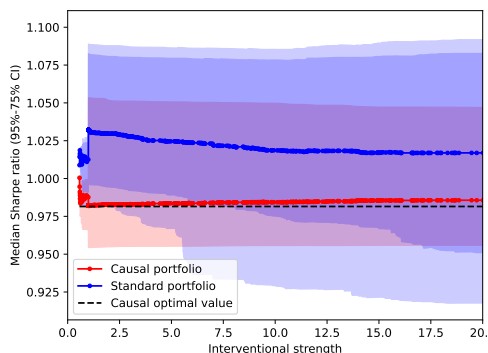

Figure 9: Worst-case Sharpe ratio vs interventional strength

Figure 10: Median and (75%-95%) CI of Sharpe ratio vs interventional strength

**Reproducibility statement.** All results can be reproduced using the source code provided in the Supplimentary Materials. Demo notebooks of the numerical experiments will be made available in a paper-related GitHub repository upon publication.

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

## A   AUXILIARY RESULTS

**Lemma A.1** (Interchangeability principle). *Let $(\Omega, \mathcal{F}, \mathbb{P})$ be a probability space and let $f : \Omega \times \mathbb{R}^d \to \overline{\mathbb{R}}$ be an $\mathcal{F}$-measurable normal integrand. Then:*

$$\int \min_{x \in \mathbb{R}^d} f(\omega, x) \mathbb{P}(d\omega) = \min_{X \in m\mathcal{F}} \int f(\omega, X(\omega)) \mathbb{P}(d\omega),$$

*provided that the right-hand side is not $\infty$.*

*Furthermore, if both sides are not $-\infty$, then:*

$$X^* \in \underset{X \in m\mathcal{F}}{\arg\min} \int f(\omega, X(\omega)) \mathbb{P}(d\omega) \Longleftrightarrow X^*(\omega) \in \underset{x \in \mathbb{R}^d}{\arg\min} f(\omega, x), \ \ (\mu\text{-almost surely})$$

*Proof.* See Rockafellar & Wets (1998, Theorem 14.60). □

**Lemma A.2** (Composition lemma). *Let $(\mathcal{X}, \|\cdot\|)$ be a Banach space with its Borel $\sigma$-algebra and let $\mu^{(0)}, \ldots, \mu^{(d)}$ be measures defined on it. Given measurable maps $\hat{T}^{(k)} : \mathcal{X} \to \mathcal{X}$ and $T^{(k)} : \mathcal{X} \to \mathcal{X}$ such that $T^{(k)}{}_\# \mu^{(k-1)} = \mu^{(k)}$ (for $k = 1, \ldots, d$), if the following two conditions hold:*

*i)* $\hat{T}^{(k)}$ *is $L_k$-Lipschitz,*

*ii)* $\|T^{(k)} - \hat{T}^{(k)}\|_{L^p(\mu^{(k-1)})} \le \varepsilon_k$,

*then:*

$$\|T^{(d)} \circ \cdots \circ T^{(1)} - \hat{T}^{(d)} \circ \cdots \circ \hat{T}^{(1)}\|_{L^p(\lambda)} \le \sum_{k=1}^d \varepsilon_k \prod_{j=k+1}^d L_j,$$

*with the convention that $\prod_{j \in \emptyset} L_j := 1$.*

*Proof.* The claim follows by induction. It is obviously true for $d = 1$. Assume that it holds for $d-1$, then for $d$:

$$\|T^{(d)} \circ \cdots \circ T^{(1)} - \hat{T}^{(d)} \circ \cdots \circ \hat{T}^{(1)}\|_{L^p(\mu^{(0)})}$$

$$\le \|T^{(d)} \circ T^{(d-1)} \circ \cdots \circ T^{(1)} - \hat{T}^{(d)} \circ T^{(d-1)} \circ \cdots \circ T^{(1)}\|_{L^p(\mu^{(0)})} +$$

$$\|\hat{T}^{(d)} \circ T^{(d-1)} \circ \cdots \circ T^{(1)} - \hat{T}^{(d)} \circ \hat{T}^{(d-1)} \circ \cdots \circ \hat{T}^{(1)}\|_{L^p(\mu^{(0)})}$$

$$\le \|T^{(d)} - \hat{T}^{(d)}\|_{L^p(\mu^{(d-1)})} + L_d \|T^{(d-1)} \circ \cdots \circ T^{(1)} - \hat{T}^{(d-1)} \circ \cdots \circ \hat{T}^{(1)}\|_{L^p(\mu^{(0)})}$$

$$\text{(Change of variable + Lipschitz)}$$

$$\le \varepsilon_d + L_d \cdot \sum_{k=1}^{d-1} \varepsilon_k \prod_{j=k+1}^{d-1} L_j \qquad\qquad \text{(claim holds of } d-1)$$

$$= \sum_{k=1}^d \varepsilon_k \prod_{j=k+1}^d L_j$$

□

**Lemma A.3.** *Let $g \in IncrMLP(n)$ with parameter space $\Theta(n)$. Then the map $\theta \mapsto g(\cdot; \theta)$ from $\Theta(n)$ to $L^1([0, 1])$ is continuous.*

*Proof.* It is a direct application of Lebesgue's dominated convergence theorem (Bogachev, 2007, Theorem 2.8.1), so we just verify that the assumptions of the theorem hold. Let $\theta_k \to \theta \in \Theta$ be any convergent sequence. Since $\theta \mapsto g(u; \theta)$ is continuous, we have that $g(u; \theta_k) \to g(u; \theta)$ for all $u \in [0,1]$. Furthermore, the functions $g(\cdot; \theta_k)$ are uniformly bounded:

$$
\begin{aligned}
\sup_{k \in \mathbb{N}} |g(u; \theta_k)| &\leq \sup_{k \in \mathbb{N}} \sup_{u \in [0,1]} |g(u; \theta_k)| \\
&\leq \sup_{k \in \mathbb{N}} \max\{|g(0; \theta_k)|, |g(1; \theta_k)|\} \qquad (u \mapsto g(u; \theta) \text{ is increasing}) \\
&\leq \sup_{\theta \in K} \max\{|g(0; \theta)|, |g(1; \theta)|\} \\
&< +\infty
\end{aligned}
$$

where $K \subseteq \Theta$ is any compact containing the sequence $\{\theta_k, k \in \mathbb{N}\}$ (which exists because the sequence is convergent) and the last inequality follows from the fact that $\theta \mapsto \max\{|g(0; \theta)|, |g(1; \theta)|\}$ is continuous (it's the minimum of two continuous functions) and therefore bounded on $K$. $\qquad\square$

**Lemma A.4.** *Let $R \subseteq \mathbb{R}^k$ be a compact set and let the functions $f(\cdot, x) : [a, b] \to \mathbb{R}$ be continuous, linear splines on a common grid $a = u_1 < \ldots < u_{n+1} = b$, for every $x \in R$. Then there exists a subset $\Theta \subseteq \Theta(n)$ (which depends only on the grid) such that the set-valued function $\tilde{\theta} : R \rightrightarrows \Theta$, defined by*

$$
\tilde{\theta}(x_{PA(k)}) := \arg\min_{\theta' \in \Theta} \|\hat{f}(\cdot, x_{PA(k)}) - g(\cdot, \theta'))\|_{L^1([0,1])}, \quad \forall x_{PA(k)} \in R
$$

*admits a continuous selection $\theta : R \to \Theta$, such that $g(u, \theta(x)) = \hat{f}(u, x)$ for all $u \in [0, 1]$.*

*Proof.* The existence of a continuous selection follows from Michael's theorem (Aubin & Frankowska, 2009, Theorem 9.1.2), provided we can show that $\tilde{\theta}$ is lower semi-continuous with closed and convex values.

Lower-semicontinuity actually holds regardless of the choice of the set $\Theta$, so we prove it first. It follows from the fact that that $(x_{PA(k)}, \theta) \mapsto \|\hat{f}(u, x_{PA(k)}) - g(u; \theta'))\|_{L^1([0,1])}$ is a Carathéodory function (for a definition, see Rockafellar & Wets (1998, Example 14.29)) and therefore a normal integrand (Rockafellar & Wets, 1998, Definition 14.27, Proposition 14.28). Indeed:

- Since $(u, x_{PA(k)}) \mapsto |\hat{f}(u, x_{PA(k)}) - g(u; \theta))|$ is measurable (even continuous) for all $\theta \in \Theta$, Tonelli's theorem (Folland, 1999, Theorem 2.37) implies that $x \mapsto \|\hat{f}(\cdot, x) - g(\cdot; \theta))\|_{L^1([0,1])}$ is measurable.

- The map $\theta \mapsto h(x_{PA(k)}, \theta)$ is continuous for all $x_{PA(k)} \in \mathbb{R}^{|PA(k)|}$ because it's the composition of two continuous maps: $\theta \mapsto g(\cdot; \theta) \in L^1([0,1])$, which is continuous by Lemma A.3, and $g(\cdot; \theta) \mapsto \|\hat{f}(\cdot, x) - g(\cdot; \theta))\|_{L^1([0,1])}$, which is continuous because the norm is a continuous function.

We will now show that $\tilde{\theta}$ is actually singleton valued (which, of course, implies that it is closed and convex valued), by constructing a suitable set $\Theta \subseteq \Theta(n)$. The main strategy is to realize that the weights and biases of the first layer ($w^{(1)}$ and $b^{(1)}$) can be used to fully specify the segments on which the function $u \mapsto g(u, \theta)$ is piecewise linear and that, once this choice is made, the weights and the bias of the second layer ($w^{(2)}$ and $b^{(2)}$) determine *uniquely* the slope and intercepts on each segment.

More specifically, given the grid $a = u_1 < u_2 < \ldots < u_{n+1} = b$, denote by

$$
\Delta u_i := u_{i+1} - u_i \quad \text{and} \quad m_i := \frac{1}{2}(u_{i+1} + u_i), \quad (i = 1, \ldots, n)
$$

the width and the midpoint of each grid segment, respectively. If we set

$$
\bar{w}_i^{(1)} = 2/\Delta u_i, \quad \bar{b}_i^{(1)} = -m_i \Delta u_i, \quad (i = 1, \ldots, n)
$$

and define $\Theta := \{\bar{w}^{(1)}\} \times \{\bar{b}^{(1)}\} \times \mathbb{R}^n_{>0} \times \mathbb{R} \subseteq \Theta(n)$, then $g(\cdot, \theta)$ is piecewise linear exactly on the grid $\{u_i\}_{i=1}^{n+1}$, for any $\theta \in \Theta$. Additionally on each segment $[u_i, u_{i+1}]$, the function $g(\cdot, \theta)$ has slope

$$w_i^{(2)} \left( \bar{w}_i^{(1)} + \frac{\alpha}{2} \sum_{j \neq i} \bar{w}_j^{(1)} \right)$$

and bias

$$w_i^{(2)}\bar{b}_i^{(1)} + nb^{(2)} + (n-1)\frac{\alpha}{2}w_j^{(2)}\bar{b}_j^{(1)} + \left(\frac{\alpha}{2} - 1\right)\left(\sum_{j<i} w_j^{(2)} - \sum_{j>i} w_j^{(2)}\right).$$

We can therefore exactly match any continuous, strictly increasing, piecewise linear function on the grid $\{u_i\}_{i=1}^{n+1}$ by matching the slope and intercept on $[u_1, u_2]$, together with the slopes on each of the remaining segments (the intercepts will be automatically matched by continuity). This is a linear system of $n+1$ equations in $n+1$ unknowns and it always admits a unique solution (as can be readily checked), which implies that for every $x \in R$ we can find a $\theta \in \Theta$ such that $g(u, \theta) = \hat{f}(u, x)$ for all $u \in [a, b]$. $\qquad \square$

**Lemma A.5.** *Let $g \in IncrMLP(n)$. Then $\theta \mapsto g(u, \theta)$ is locally Lipschitz uniformly in $u \in [0, 1]$, i.e. for every compact $K \subseteq \Theta(n)$ there exists an $L > 0$ such that:*

$$|g(u, \theta) - g(u, \theta')| \leq L\|\theta - \hat{\theta}\|, \quad \forall \theta, \hat{\theta} \in K, \ \forall u \in [0, 1].$$

*Proof.* The proof follows by direct computation. We use repeatedly the Cauchy-Schwartz inequality, the fact that the activation $\rho$ is 1-Lipschitz and that $\|u\| \leq 1$:

$$\begin{aligned}
|g(u, \theta) - g(u, \hat{\theta})| \leq &|\langle w^{(2)}, \rho^{\otimes n}(uw^{(1)} + b^{(1)})\rangle + b^{(2)} - \langle \hat{w}^{(2)}, \rho^{\otimes n}(u\hat{w}^{(1)} + \hat{b}^{(1)})\rangle - \hat{b}^{(2)}| \\
\leq &|\langle w^{(2)}, \rho^{\otimes n}(uw^{(1)} + b^{(1)})\rangle - \langle \hat{w}^{(2)}, \rho^{\otimes n}(uw^{(1)} + b^{(1)})\rangle| \\
&+ |\langle \hat{w}^{(2)}, \rho^{\otimes n}(uw^{(1)} + b^{(1)})\rangle - \langle \hat{w}^{(2)}, \rho^{\otimes n}(u\hat{w}^{(1)} + \hat{b}^{(1)})\rangle| + |b^{(2)} - \hat{b}^{(2)}| \\
= &|\langle w^{(2)} - \hat{w}^{(2)}, \rho^{\otimes n}(uw^{(1)} + b^{(1)})\rangle| \\
&+ |\langle \hat{w}^{(2)}, \rho^{\otimes n}(uw^{(1)} + b^{(1)}) - \rho^{\otimes n}(u\hat{w}^{(1)} + \hat{b}^{(1)})\rangle| + |b^{(2)} - \hat{b}^{(2)}| \\
\leq &\|w^{(2)} - \hat{w}^{(2)}\|\|\rho^{\otimes n}(uw^{(1)} + b^{(1)})\| \\
&+ \|\hat{w}^{(2)}\|\|\rho^{\otimes n}(uw^{(1)} + b^{(1)}) - \rho^{\otimes n}(u\hat{w}^{(1)} + \hat{b}^{(1)})\| + |b^{(2)} - \hat{b}^{(2)}| \\
\leq &\|w^{(2)} - \hat{w}^{(2)}\|(\|w^{(1)}\| + \|b^{(1)}\|)) + \|\hat{w}^{(2)}\|(\|w^{(1)} - \hat{w}^{(1)}\| + \|b^{(1)} - \hat{b}^{(1)}\| + |b^{(2)} - \hat{b}^{(2)}|)
\end{aligned}$$

Since the parameters are contained in a compact $K$, their norms are bounded by a constant, say $M > 0$, so that:

$$\begin{aligned}
|g(u, \theta) - g(u, \hat{\theta})| &\leq 2M(\|w^{(2)} - \hat{w}^{(2)}\| + \|w^{(1)} - \hat{w}^{(1)}\| + \|b^{(1)} - \hat{b}^{(1)}\| + |b^{(2)} - \hat{b}^{(2)}|) \\
&\leq 2M\sqrt{4}\|\theta - \hat{\theta}\|
\end{aligned}$$

where the last inequality is due to Cauchy-Schwartz (this time applied to the (four-dimensional) vector of parameters' norms and the four-dimensional unit vector). $\qquad \square$

**Lemma A.6.** *Let $(u, x_{PA}(k)) \mapsto F_k^{-1}(u \mid x_{PA(k)})$ be as in Theorem 4.4. Then:*

*i)* $F_k^{-1}(u \mid x_{PA(k)}) \in L^1(du \otimes \mu(dx_{PA(k)}))$,

*ii)* $\partial_u F_k^{-1}(u \mid x_{PA(k)}) \in L^1(du \otimes \mu(dx_{PA(k)}))$,

*iii)* $\partial_{x_j} F_k^{-1}(u \mid x_{PA(k)}) \in L^1(du \otimes \mu(dx_{PA(k)}))$, *for all $j \in PA(k)$.*

*Proof.*  i) By direct integration:

$$
\int_{\mathbb{R}^{|\mathrm{PA}(k)|}} \mu(dx_{\mathrm{PA}(k)}) \int_{[0,1]} du |F_k^{-1}(u \mid x_{\mathrm{PA}(k)})|
$$

$$
= \int_{\mathbb{R}^{|\mathrm{PA}(k)|}} \mu(dx_{\mathrm{PA}(k)}) \int_{\mathbb{R}} \mu(dx_k \mid x_{\mathrm{PA}(k)})|x_k|
$$

$$
= \int_{\mathbb{R}} \mu(dx_k)|x_k| \leq +\infty
$$

where we have first used the change-of-variable formula (Bogachev, 2007, Theorem 3.6.1) with $F_k^{-1}(\cdot \mid x_{\mathrm{PA}(k)})_{\#}\mathcal{U}[0,1] = \mu(dx_k \mid x_{\mathrm{PA}(k)})$ (McNeil et al., 2015, Proposition A.6) and then used the fact that $\mu$ has finite first moments.

ii) $u \mapsto F^{-1}(u \mid x)$ is increasing on the closed interval $[0,1]$, therefore by Bogachev (2007, Corollary 5.2.7) it is almost everywhere differentiable and:

$$
\int_{[0,1]} |\partial_u F_k^{-1}(u \mid x_{\mathrm{PA}(k)})| du \leq F_k^{-1}(1 \mid x_{\mathrm{PA}(k)}) - F_k^{-1}(0 \mid x_{\mathrm{PA}(k)}).
$$

The right-hand side is just $\mathrm{diam}(\mathrm{supp}(\mu(dx_k \mid x_{\mathrm{PA}(k)})))$, which is finite, since $\mu$ is compactly supported.

iii) Continuity of $u \mapsto F_k(u \mid x_{\mathrm{PA}(k)})$ implies that $F_k(F_k^{-1}(u \mid x_{\mathrm{PA}(k)})) = u$ (McNeil et al., 2015, Proposition A.3 (viii)). Differentiating this expression on both sides and using the chain rule yields:

$$
\int_{[0,1]} \left| \partial_{x_j} F_k^{-1}(u \mid x_{\mathrm{PA}(k)}) \right| du = \int_{[0,1]} du \left| -\frac{\partial_{x_j} F_k(F_k^{-1}(u \mid x_{\mathrm{PA}(k)}) \mid x_{\mathrm{PA}(k)})}{\partial_u F_k(F_k^{-1}(u \mid x_{\mathrm{PA}(k)}) \mid x_{\mathrm{PA}(k)})} \right|
$$

$$
= \int_{\mathbb{R}} dx' \left| -\partial_{x_j} F_k(x' \mid x_{\mathrm{PA}(k)}) \right|,
$$

where the second equality follows from the same change-of-variable as in part (i) and by simplifying the conditional density. The claim now follows by integrating over $\mathbb{R}^{\mathrm{PA}(k)}$ with respect to $\mu(dx_{\mathrm{PA}(k)})$ and using the assumption that $(x_k, x_{\mathrm{PA}(k)}) \mapsto F_k(x_k \mid x_{\mathrm{PA}(k)})$ is a $C^1$ map and therefore admits bounded partial derivatives on compacts.

$\square$

# B  PROOFS

## B.1  PROOF OF THEOREM 3.7

*Proof.* We generalize the proof by Acciaio et al. (2024) to our $G$-causal setting. Given $\mu, \nu \in \mathcal{P}_G(\mathbb{R}^d)$, let $g$ be a $G$-causal function and let $\pi$ be the optimal $G$-bicausal coupling between $\mu$ and $\nu$. Then:

$$
-\mathbb{E}^\nu \left[ Q(X_T, g(X_{V \setminus T})) \right] = - \int Q(x'_T, g(x'_{V \setminus T}))\nu(dx')
$$

$$
= - \int Q(x'_T, g(x'_{V \setminus T}))\pi(dx, dx')
$$

$$
= \int \left( Q(x_T, g(x'_{V \setminus T})) - Q(x'_T, g(x'_{V \setminus T})) \right) \pi(dx, dx') - \int Q(x_T, g(x'_{V \setminus T}))\pi(dx, dx')
$$

Since $x \mapsto Q(x, g)$ is uniformly locally $L$-Lipschitz, the first integral satisfies:

$$\int \Big( Q(x_T, g(x'_{V \setminus T})) - Q(x'_T, g(x'_{V \setminus T})) \Big) \pi(dx, dx') \leq L \int \|x_T - x'_T\| \pi(dx, dx')$$

$$\leq L \int \|x - x'\| \pi(dx, dx')$$

$$= L \cdot W_G(\mu, \nu)$$

For the second integral, we notice that:

$$- \int Q(x_T, g(x'_{V \setminus T})) \pi(dx, dx') \leq - \int Q\Big( x_T, \int g(x'_{V \setminus T}) \pi(dx' \mid x) \Big) \mu(dx)$$

$$= - \int Q\Big( x_T, \underbrace{\int g(x'_{\mathrm{PA}(T)}) \pi(dx' \mid x)}_{h(x)} \Big) \mu(dx)$$

where we first applied Jensen's inequality and then the fact that $g$ is $G$-causal. Furthermore, since $\pi$ is $G$-causal, the function $h(x) := \int g(x'_{V \setminus T}) \pi(dx' \mid x)$ actually depends only on $x_{\mathrm{PA}(T) \cup \mathrm{PA}(\mathrm{PA}(T))}$. To ease the notation, denote $A := \mathrm{PA}(T) \cup \mathrm{PA}(\mathrm{PA}(T))$. Then:

$$- \int Q\left( x_T, h(x_A) \right) \mu(dx) = - \int \mu(dx_A) \int \mu(dx_T \mid x_A) Q(x_T, h(x_A))$$

$$= - \int \mu(dx_A) \int \mu(dx_T \mid x_{\mathrm{PA}(T)}) Q(x_T, h(x_A))$$

$$\leq - \int \mu(dx_A) \int \mu(dx_T \mid x_{\mathrm{PA}(T)}) Q(x_T, h^*(x_{\mathrm{PA}(T)}))$$

$$= -\mathcal{V}(\mu)$$

where in the second equality we have used the fact that $X_T \perp\!\!\!\perp X_A \mid X_{\mathrm{PA}(T)}$ (see condition (ii) in Definition 2.2 or simply notice that $X_{\mathrm{PA}(T)}$ $d$-separates $X_T$ and $X_{\mathrm{PA}(\mathrm{PA}(T))}$), while the inequality is due to Eq. (1).

Putting everything together:

$$-\mathbb{E}^\nu \left[ Q(Y, g(X)) \right] \leq L \cdot W_G(\mu, \nu) - \mathcal{V}(\mu)$$

and, since $g$ is arbitrary, we obtain:

$$\mathcal{V}(\mu) - \mathcal{V}(\nu) \leq L \cdot W_G(\mu, \nu).$$

By symmetry, exchanging $\mu$ and $\nu$ yields the same inequality for the term $\mathcal{V}(\nu) - \mathcal{V}(\mu)$, therefore

$$|\mathcal{V}(\mu) - \mathcal{V}(\nu)| \leq L \cdot W_G(\mu, \nu).$$

$\square$

### B.2 Proof of Theorem 4.4

*Proof.* We know that $\mu = T_\# \mathcal{N}(0, I_d)$, where $T = T^{(d)} \circ \cdots \circ T^{(1)}$ is the $G$-compatible, increasing transformation in the statement of Theorem 4.3. Now, let $\hat{T} = \hat{T}^{(d)} \circ \cdots \circ \hat{T}^{(1)} \in G\text{-NF}(d)$ be a G-NFwith flows as in Eq. (2) and define the $G$-bicausal coupling $\pi := (T, \hat{T})_\# \mathcal{N}(0, 1)$, then we have that:

$$W_G(\mu, \hat{T}_\# \lambda) \leq \int_{\mathbb{R}^d \times \mathbb{R}^d} \|x - x'\| \pi(dx, dx') = \int_{[0,1]^d} \|T(u) - \hat{T}(u)\| du.$$

We can make the right-hand side smaller than any $\varepsilon > 0$ by using Lemma A.2 (with $\mathcal{X} := [0, 1]^d$, $\mu^{(0)} := \mathcal{U}([0, 1]^d)$ and $\mu^{(k)} := \mu_{1:k} \otimes \mathcal{U}([0, 1]^{d-k})$, for $k = 1, \ldots, d$), provided that we can show that conditions (i) and (ii) therein hold.

**Condition (i).** Each hypercoupling flow $\hat{T}^{(k)}$ differs from the identity only at its $k$-th coordinate, which is the output of a shallow MLP (see Eq. (2)) . But shallow MLPs are Lipschitz functions of their input, therefore each $\hat{T}^{(k)}$ is a Lipschitz function.

**Condition (ii).** We need to show that for every $\varepsilon > 0$, there exists an $n \in \mathbb{N}$, a $\hat{\theta}(\cdot) \in$ MLP and a $g(\cdot, \hat{\theta}(x_{\mathrm{PA}(k)})) \in \mathrm{IncrMLP}(n)$ such that

$$\int_{[0,1]} du \int_{\mathbb{R}^{|\mathrm{PA}(k)|}} \mu(dx_{\mathrm{PA}(k)})|F_k^{-1}(u \mid x_{\mathrm{PA}(k)}) - g(u; \hat{\theta}(x_{\mathrm{PA}(k)}))| \leq \varepsilon. \tag{7}$$

We will prove this bound by splitting the error into three terms and bounding each one separately.

**Term 1.** First we approximate $(u, x_{\mathrm{PA}}(k)) \mapsto F_k^{-1}(u \mid x_{\mathrm{PA}(k)})$ with a continuous tensor-product linear spline, $\hat{f}(u, x_{\mathrm{PA}(k)})$, on the rectangle $[0,1] \times R$, where $R = \prod_{j=1}^{|\mathrm{PA}(k)|}[a_j, b_j]$ is a rectangle large enough to contain the compact support of $\mu(dx_{\mathrm{PA}(k)})$. We choose the approximation grid fine enough to satisfy:

$$\int_{[0,1]} du \int_{\mathbb{R}^{|\mathrm{PA}(k)|}} \mu(dx_{\mathrm{PA}(k)})|F_k^{-1}(u \mid x_{\mathrm{PA}(k)}) - \hat{f}(u, x_{\mathrm{PA}(k)})| \leq \varepsilon/2,$$

and let $n + 1$ be the number of gridpoints in the $u$-axis (i.e. the grid on $[0,1]$ has gridpoints $0 = u_1 < \ldots < u_{n+1} = 1$).

The validity of this approximation follows from (Schumaker, 2007, Theorem 12.7) and requires that $(u, x) \mapsto F^{-1}(u|x)$ belong to a suitable tensor Sobolev space (Schumaker, 2007, Example 13.5), as we verify in Lemma A.6.

**Term 2.** Next, we approximate the univariate functions $u \mapsto \hat{f}(u, x_{\mathrm{PA}(k)})$, for each $x_{\mathrm{PA}(k)} \in R$, with neural networks $g(\cdot; \theta(x_{\mathrm{PA}(k)})) \in \mathrm{IncrMLP}(n)$, by judiciously choosing the function $\theta : R \to \Theta(n)$.

Since all the functions $\hat{f}(\cdot, x_{\mathrm{PA}(k)})$ share the same grid on $[0, 1]$, by Lemma A.4 there exists a parameter subset $\Theta \subseteq \Theta(n)$ (which depends only on this grid) such that the set-valued map $\tilde{\theta} : R \rightrightarrows \Theta$, defined as

$$\tilde{\theta}(x_{\mathrm{PA}(k)}) := \underset{\theta' \in \Theta}{\arg\min} \ \|\hat{f}(\cdot, x_{\mathrm{PA}(k)}) - g(\cdot, \theta'))\|_{L^1([0,1])},$$

admits a continuous selection $\theta : R \to \Theta$. We then use this function $\theta$ to parametrize the neural networks $g(\cdot, \theta(x_{\mathrm{PA}(k)}))$ and, as implied by Lemma A.4, this parametrization is optimal, in the sense that $g(u, \theta(x_{\mathrm{PA}(k)})) = \hat{f}(u, x_{\mathrm{PA}(k)})$ for all $u \in [0, 1]$, thus achieving zero approximation zero, i.e.

$$\int_{[0,1]} du \int_{\mathbb{R}^{|\mathrm{PA}(k)|}} \mu(dx_{\mathrm{PA}(k)})|\hat{f}(u, x_{\mathrm{PA}(k)}) - g(u, \theta(x_{\mathrm{PA}(k)}))| = 0.$$

**Term 3.** Finally, we approximate $g(u; \theta(x_{\mathrm{PA}(k)}))$ with $g(u; \hat{\theta}(x_{\mathrm{PA}(k)}))$, where $\hat{\theta}(\cdot)$ is a suitable MLP.

Since $\theta : R \to \Theta$ is a continuous function on a compact, we have that $\theta \in L^1(\mu)$, therefore for every $\varepsilon' > 0$ there is an MLP[7] $\hat{\theta}$ such that $\|\theta - \hat{\theta}\|_{L^1(\mu)} \leq \varepsilon'$ (Leshno et al., 1993, Proposition 1).

Therefore:

$$\int_{[0,1]} du \int_{\mathbb{R}^{|\mathrm{PA}(k)|}} \mu(dx_{\mathrm{PA}(k)})|g(u; \theta(x_{\mathrm{PA}(k)})) - g(u; \hat{\theta}(x_{\mathrm{PA}(k)}))|$$

$$\leq \int_{[0,1]} du \int_{\mathbb{R}^{|\mathrm{PA}(k)|}} \mu(dx_{\mathrm{PA}(k)})L \ \|\theta(x_{\mathrm{PA}(k)}) - \hat{\theta}(x_{\mathrm{PA}(k)})\|$$

$$\leq L\varepsilon' \leq \varepsilon/2 \qquad\qquad (\text{choose } \varepsilon' = \varepsilon/2L)$$

where the first inequality follows from the uniform local Lipschitz property on the compact $\theta(\mathrm{supp}(\mu)) \cup \hat{\theta}(\mathrm{supp}(\mu))$ proved in Lemma A.5.

Summing all three approximation errors together, we obtain the bound in Eq. (7). □

---

[7]For the theorem to hold we only need the activation function to be non-polynomial and locally essentially bounded (such as ReLU).

### B.3 PROOF OF THEOREM 4.6

*Proof.* First we notice that

$$W_G(\mu, \nu) = \min_{\pi \in \Pi_G^{\text{bc}}(\mu, \nu)} \int_{\mathbb{R}^d \times \mathbb{R}^d} \|x - x'\| \, \pi(dx, dx')$$

$$\leq \min_{\pi \in \Pi_G^{\text{bc}}(\mu, \nu)} \int_{\mathbb{R}^d \times \mathbb{R}^d} \text{diam}(K) \cdot \mathbf{1}_{\{x \neq x'\}}, \pi(dx, dx')$$

$$=: \text{diam}(K) \cdot d_{G\text{-}TV}(\mu, \nu)$$

where in the last equality we have introduced the $G$-causal total variation distance, $d_{G\text{-}TV}(\cdot, \cdot)$, as a suitable generalization of the total variation distance for $G$-bicausal couplings.

The claim then follows by showing that $d_{G\text{-}TV}(\mu, \nu) \leq (2^d - 1)d_{TV}(\mu, \nu)$ for all sorted DAGs $G$ by induction on the number of vertices, which is a straightfoward but tedious generalization of Eckstein & Pammer (2024, Lemma 3.5) to our $G$-causal setting.

The claim holds trivially if $G$ has only one vertex (all couplings are $G$-bicausal). Suppose now the claim is true for all sorted DAGs on $n$ vertices. Then for a sorted DAG $G$ on $n + 1$ vertices, denote by $G_n$ its subgraph on vertices $\{1, \ldots, n\}$. We start with some definitions. Define:

$$\eta(dx_{n+1}|x_{\text{PA}(n+1)}) := \mu(dx_{n+1}|x_{\text{PA}(n+1)}) \wedge \nu(dx_{n+1}|x_{\text{PA}(n+1)}),$$

$$\pi \in \Pi_G^{\text{bc}}(\mu, \nu) \text{ as } \pi := \pi_n \otimes \pi(dx_{n+1}, dx'_{n+1} \mid x_{\text{PA}(n+1)}, x'_{\text{PA}(n+1)}),$$

where $\pi_n \in \Pi_{G_n}^{\text{bc}}(\mu(dx_{1:n}), \nu(dx'_{1:n}))$, and:

$$\pi(dx_{n+1}, dx'_{n+1}|x_{\text{PA}(n+1)}, x'_{\text{PA}(n+1)}) := \begin{cases} \sigma(dx_{n+1}, dx'_{n+1}|x_{\text{PA}(n+1)}, x'_{\text{PA}(n+1)}) & \text{if } x_{\text{PA}(n+1)} = x'_{\text{PA}(n+1)} \\ \mu(dx_{n+1} \mid x_{\text{PA}(n+1)}) \otimes \nu(d'x_{n+1} \mid x'_{\text{PA}(n+1)}) & \text{otherwise} \end{cases}$$

where $\sigma$ is the optimal coupling for the (conditional) total variation distance, i.e.:

$$\sigma(dx_{n+1}, dx'_{n+1}|x_{\text{PA}(n+1)}, x'_{\text{PA}(n+1)}) := (\text{id}, \text{id})_\# \eta(dx_{n+1}|x_{\text{PA}(n+1)}) + (\mu(dx_{n+1}|x_{\text{PA}(n+1)})$$
$$- \eta(dx_{n+1}|x_{\text{PA}(n+1)})) \otimes (\nu(dx_{n+1} \mid x_{\text{PA}(n+1)}) - \eta(dx_{n+1}|x_{\text{PA}(n+1)}))$$

Then the following bounds can be established (see Eckstein & Pammer (2024, Lemma 3.5) for step-by-step details):

$$d_{G\text{-}TV}(\mu, \nu) \leq \int \mathbf{1}_{\{x \neq x'\}} \pi(dx, dx')$$

$$= \int \mathbf{1}_{\{x_{1:n} \neq x'_{1:n}\}} \pi_n(dx_{1:n}, dx'_{1:n})$$

$$+ \int d_{TV}(\mu(dx_{n+1}|x_{\text{PA}(n+1)}), \nu(dx_{n+1}|x_{\text{PA}(n+1)})) \mathbf{1}_{\{x_{1:n} = x'_{1:n}\}} \pi_n(dx_{1:n}, dx'_{1:n})$$

$$= \int \mathbf{1}_{\{x_{1:n} \neq x'_{1:n}\}} \pi_n(dx_{1:n}, dx'_{1:n}) + \|\eta \otimes (\mu(dx_{n+1}|x_{\text{PA}(n+1)}) - \nu(dx_{n+1}|x_{\text{PA}(n+1)}))\|_{TV}$$

For all $A \in \mathbb{R}^{n+1}$, one has:

$$\eta \otimes (\mu(dx_{n+1}|x_{\text{PA}(n+1)}) - \nu(dx_{n+1}|x_{\text{PA}(n+1)}))(A) \leq \|\mu(dx_{n+1}|x_{\text{PA}(n+1)}) - \nu(dx_{n+1}|x_{\text{PA}(n+1)})\|_{TV}$$

$$+ \int \mathbf{1}_{\{x_{1:n} \neq x'_{1:n}\}} \pi_n(dx_{1:n}, dx'_{1:n})$$

Putting the two bounds together and minimizing over all $G_n$-bicausal couplings $\pi_n$:

$$d_{G\text{-}TV}(\mu, \nu) \leq 2d_{G_n\text{-}TV}(\mu_{1:n}, \nu_{1:n}) + d_{TV}(\mu, \nu)$$
$$\leq (2^{n+1} - 2 + 1)d_{TV}(\mu, \nu)$$
$$= (2^{n+1} - 1)d_{TV}(\mu, \nu)$$

where we have used:

$$d_{G_n\text{-}TV}(\mu_{1:n}, \nu_{1:n}) \leq (2^n - 1)d_{TV}(\mu_{1:n}, \nu_{1:n}) \leq (2^n - 1)d_{TV}(\mu, \nu),$$

which follows from the induction hypothesis and the data pre-processing inequality for the total variation distance (Eckstein & Nutz, 2022, Lemma 4.1). $\square$

