# OpenReview forum: "Robust Optimization in Causal Models and $G$-Causal Normalizing Flows"
_ICLR.cc/2026/Conference — ICLR 2026 Conference Withdrawn Submission_

### Official Review · Reviewer_T2by · 2025-10-18

**Soundness:** 3
**Presentation:** 4
**Contribution:** 2
**Rating:** 2
**Confidence:** 4

**Summary:**

This paper studies the causal optimization problems. It proves that optimization problems respecting a given causal structure (G-causal optimization problems) are continuous under a specific metric called the G-causal Wasserstein distance ($W_G$). It also proves that solutions to these G-causal optimization problems are inherently interventionally robust. It introduces the G-causal normalizing flow (G-NF), a novel generative model architecture that explicitly incorporates the causal Directed Acyclic Graph. The authors prove this model has a universal approximation property for causal models and that it can be trained to minimize the G-causal Wasserstein distance using standard maximum likelihood estimation.

**Strengths:**

1. The authors show the discontinuity of causal optimization problems under standard metrics (Example 3.8) and prove their continuity under the G-causal Wasserstein distance (Theorem 3.7).

2. Universal Approximation Property is proven for causal normal flow. In particular, the authors explicitly construct the causal normal flow, which provides guidelines for practice.

3. The authors verify their proposed method with synthetic data.

**Weaknesses:**

1. The practical motivation for the G-causal problem is unclear to me. The authors do not mention any real-world application of this problem in their paper.

2. The optimization problem is optimized over ANY functions from $R^{|V\backslash T|}$ to $R^{|T|} $. However, in practice, we usually impose some smoothness assumption on $h$ or the functions in the structure equations, e.g. Lipschitzness, differentiability. In particular, with these extra structures, the problem may be continuous. For example, [1] shows that if $f_i$ is Lipschitz continuous, the ATE is continuous in the standard Wasserstein distance.

3. The G-causal function requires that $h$ should only depend on $Pa(T)$. However, in causal inference, we usually care about the descendants of T [2]. Is it possible to generalize your results to this case?


[1] Tan, Jiyuan, Jose Blanchet, and Vasilis Syrgkanis. "Consistency of neural causal partial identification." Advances in Neural Information Processing Systems 37 (2024): 68956-68999.
[2] Xia, Kevin, et al. "The causal-neural connection: Expressiveness, learnability, and inference." Advances in Neural Information Processing Systems 34 (2021): 10823-10836.

**Questions:**

Please address my concern in the weakness part.

---

### Official Review · Reviewer_nViu · 2025-10-28

**Soundness:** 4
**Presentation:** 3
**Contribution:** 3
**Rating:** 6
**Confidence:** 4

**Summary:**

This paper addresses the problem of optimization in causal models under potential distributional shifts (interventions). The authors' core contribution is twofold. First, they provide a key theoretical insight: they show that interventionally robust optimization problems in causal models are continuous under the "G-causal Wasserstein distance" ($W_G$) but may be discontinuous under the standard Wasserstein distance. This motivates their second contribution: a novel generative model, the "G-causal Normalizing Flow" (G-NF), which is an invertible architecture that explicitly respects a given causal DAG. The authors prove this model has a universal approximation property w.r.t. $W_G$ and that standard maximum likelihood training (KL minimization) provably minimizes this distance. Finally, they demonstrate empirically that using G-NF for data augmentation in causal regression and portfolio optimization tasks leads to downstream optimizers that are significantly more robust to interventions compared to those trained on data from standard (non-causal) generative models like VAEs and RealNVP.

**Strengths:**

1. **Soundness**: The paper is well-written, well-organized, and easy to follow. The logical progression from problem motivation to theory, and then from theory to method and experiments, is exemplary.

2. **Novelty**: The paper formally establishes a continuity result for optimization over causal models under the G-causal Wasserstein distance, extending earlier work from adapted and time-causal optimal transport to arbitrary DAGs. The introduction of G-causal normalizing flows as a universal approximator under mild assumptions is novel and represents a substantial technical advancement over standard normalizing flow approaches.


3. **Significance / Impact**: By refining the notion of robustness in causal models, the work has immediate implications for high-impact domains (e.g., financial modeling, robust learning under intervention) and sets the stage for further research in causal generative modeling and robust optimization.

4. **Technical Soundness**: Theoretically, the core results are well-motivated, with proofs for continuity, universal approximation, and training guarantees provided. Experimental evaluation includes comprehensive empirical results contrasting causal and non-causal models, using realistic intervention scenarios and reporting on relevant metrics.

5. **Clarity**: The paper is generally well-organized, with explicit notation, figures clarifying experimental setup, and a clear distinction between theoretical and empirical sections. Proof sketches and references are provided in the main body and appendices, supporting the claims.

**Weaknesses:**

1. The entire method, both the G-causal optimizer (Def 3.4) and the G-causal normalizing flow (Section 4), fundamentally relies on having access to the correct, sorted causal DAG $G$ as an input. The paper briefly mentions causal discovery methods but does not investigate the sensitivity of the model to errors in this graph.

2. Theorem 3.5 defines robustness against interventions that alter mechanisms but preserve the conditional law of the target given its parents, and that keep the support of parent variables within the training support. This excludes interventions that shift parents outside observed support or directly tamper with the target’s mechanism. The paper claims “always interventionally robust,” but this is only within that restricted family $\mathcal{I}(\mu)$.

3. In the augmentation experiments, G-causal flows are compared mainly to standard non-causal models (RealNVP, VAE). However, there is an emerging line of causal / adapted / time-causal generative models (e.g. causal OT GANs, time-causal VAEs) that also try to align the generator with causality or temporal adaptation.

4. The regression and intervention experiments are done on randomly generated linear Gaussian SCMs, and the portfolio example is also synthetic. While this is a reasonable first step for a theory-heavy paper, the lack of any real-world or semi-realistic benchmark weakens the empirical significance claim.

5. The paper reports worst-case and median performance vs interventional strength, but does not show sensitivity to key design knobs such as:

 - architecture depth/width of the hypernetworks that parameterize the monotone couplings (θ(·) in Eq. (2))
  - sample size for augmentation (does robustness still hold with fewer than 10k synthetic samples?);
  - risk aversion $\gamma$ in portfolio optimization.Without these, it’s difficult to judge stability.
6. The universal approximation result (Theorem 4.4) assumes absolute continuity, compact support, and $C^1$ conditional CDFs. The KL→$W_G$ control (Theorem 4.6) assumes both distributions lie in a compact set $K$. These are mathematically reasonable but practically restrictive.

7. Section 5 references supplementary code and notebooks, but the main paper does not report training hyperparameters, network sizes, optimizer settings, or computational budget for the G-causal normalizing flows. That makes it harder for an expert reader to reproduce the figures without the supplement.


## Minor

Please correct below typos:

- "modelling" (line 053) → modeling

- quotiened " (line 142) -> quotiented

- " PA(Y) " (line 205) -> PA(T)

- " PA(Y) " (line 208) -> PA(T)

- "Supplimentary" (line 478) → Supplementary

**Questions:**

1. My main practical concern is the reliance on a correct causal DAG $G$. Let $G$ be the estimated DAG (used for defining the G-causal optimizer and training the G-NF) and $G^\ast$ be the true causal DAG. Is the solution of a G-causal optimizer (defined w.r.t. $G$) still robust w.r.t. the true distribution $P^\ast \in \mathcal{P}\_{G^\ast}$ if $G \neq G^\ast$? Does the robustness property (Theorem 3.5) break down completely, or is the performance degradation graceful?

2. Have you tried re-running Figures 3–8 with a perturbed graph to see whether the claimed robustness persists or collapses?


3. The architecture in Eq. (2)–(4) uses monotone scalar couplings whose parameters are produced by a hypernetwork $\theta(x_{\mathrm{PA}(i)})$.


- What is the dimensionality / depth of these hypernetworks in your experiments?
- How do you enforce strict monotonicity in practice (positivity constraints on certain weights, etc.) and how numerically stable is the inversion of $g$ during likelihood training?
- What optimizer / batch size / number of epochs did you use?

4. Theorem 4.6 states $W_G(\mu,\nu) \le C \sqrt{\tfrac{1}{2} \mathrm{DKL}(\mu | \nu)}$ under compact support.


- Can you provide more intuition for the constant $C$?
- Is $C$ data-dependent (e.g. diameter of support), and if so, how large is it in your numerical settings?
- Do you observe an empirical correlation between log-likelihood and $W_G$ distance in practice?

5. The counterexample in 3.8 is very effective at showing the discontinuity of $\mathcal{V}(\mu_\epsilon)$ under the standard Wasserstein distance. As a complement, is it possible to show (even just empirically) that $\mathcal{V}(\mu_\epsilon)$ *is* continuous under $W_G$ as $\epsilon \to 0$? This would nicely complete the theoretical motivation.

## Suggested Improvements

1. Add an experiment where G is perturbed (extra/missing edges), and report how much the worst-case performance and continuity guarantees degrade.

2. Include, or at least discuss, baselines that optimize adapted Wasserstein / causal OT distances, not just likelihood under an unconditional flow.

3. Add at least one experiment on a real dataset with a commonly used causal graph or factor model, even if approximate, to show applied relevance.

---

### Official Review · Reviewer_gQUB · 2025-11-02

**Soundness:** 3
**Presentation:** 1
**Contribution:** 2
**Rating:** 2
**Confidence:** 4

**Summary:**

This work studies the importance of having of causal generative models when used for data augmentation tasks. To this end, the authors define a $G$-causal Wassserstein distance, and show that interventionally robust optimization problems are continuous under this semi-metric but not the usual Wasserstein distance. Then, the authors propose an architecture to build G-causal normalizing flows and theoretically show that universal approximantors and can be trained using MLE. Finally, the authors demonstrate the value of their construction on a synthetic causal-regression experiment and a conditional mean-variance portfolio optimization problem.

**Strengths:**

- **S1.** I find the main finding quite compelling: that interventionally robust problems are continuous under a $G$-aware metric, but not under unaware ones.
- **S2.** I similarly appreciate the clarity on the exposition of the main content, as most details and proofs are actually given in the main pages.
- **S3.** Theorem 4.6 feels extremely interesting to me, and justifies the usage of MLE to train these models (yet I should clarify that I have not checked the proof).
- **S4.**  While it is unclear to me why not sticking to existing normalizing flows (NFs) architectures, the activation function used in Eq. 4 is interesting.

**Weaknesses:**

- **W1.** One core issue of this submission is the lack of references to existing causal generative models, e.g.: [NCM](https://arxiv.org/abs/2107.00793), [VACA](https://arxiv.org/abs/2110.14690), or [DCM](http://arxiv.org/abs/2302.00860). Especially problematic is the lack of references to [Causal Normalizing Flows](https://arxiv.org/abs/2306.05415), which defined normalizing flows that are $G$-causal as defined in this work, are trained with MLE as well, and which shown identifiability and universal approximation capabilities of the models. As a result, this casts doubts on the novelty of the proposed model and some of the theoretical results. For example, theorem 4.3 is known and indeed the transformation is unique and the transformation in the proof is known as the Darmois construction employed before in [NFs](http://arxiv.org/abs/1912.02762) and [ICA](https://linkinghub.elsevier.com/retrieve/pii/S0893608098001403).
- **W2.** The proposed coupling architecture, while functional, it is for all purposes a sequential model where there is no parameter amortization (leaving aside the quotiened variables). That is, all layers could be trained in parallel as each variable is processed sequentially one after the other.
- **W3.** While the paper is mostly well-written, the lack of citations are egregious. For an example, the paper talks and defines coupling NFs, uses RealNVP, but there is no mention to the original papers by Laurent Dinh introducing both of these.
- **W4.** The experiments need some improvements and rework. Besides the lack of baselines, most of the space is devoted to compare two regressors w/ and w/o non-parent nodes, figures show redundant information, and while the experiments show positive results, they do not seem significantly that much better than the simplest baseline (e.g. is semantically meaningful a ratio of 0.925 vs one of 0.950 in Fig. 10?).

**Questions:**

- Q1. Is the set of $G$-bicausal couplings between two measures always non-empty? (re Def. 2.4)
- Q2. Where does that the definition of distributionally robust in Thrm. 3.5 come from? And, more generally, which definitions are and are not new?
- Q3. Can you explain in more detail the inequality in the proof in Thrm. 3.5?
- Q4. Is my understanding correct, and figures 3 to 6 compare a regressor with all variables versus one with only the parents of the outcome variable? That is, with no data augmentation?

----

Other feedback:
- It is confusing to use $T$ for the normalizing flow layers and target variables.

---

### Official Review · Reviewer_aPUi · 2025-11-08

**Soundness:** 3
**Presentation:** 3
**Contribution:** 2
**Rating:** 2
**Confidence:** 3

**Summary:**

This paper introduces an interesting and novel insight that interventionally robust optimization problems, i.e., those where causal parents are used to predict a set of target features, are continuous under the G-causal Wasserstein distance.
The authors leverage this insight for a data augmentation task, demonstrating how the proposed framework can be applied beyond theoretical analysis.  Finally, they provide experimental results on synthetic datasets under various setups to support their theoretical findings.

**Strengths:**

The paper is clearly written and easy to follow, with an intuitive presentation of ideas.  Training G-causal normalizing flows by optimizing the G-causal Wasserstein distance is a novel and interesting contribution.  The authors also provide detailed experimental results that support their claims and demonstrate the effectiveness of the proposed approach.

**Weaknesses:**

Below I provide my comments.



## Major:
1. The authors propose a normalizing-flow-based approach to train a predictor for a set of target variables from their parents (as in Theorem 3.5). They also present experimental results to illustrate these capabilities in Section 5.1. However, to my knowledge, this is a well-known result, i.e., if we learn a predictor using only causal features (parents), that predictor should remain invariant across domains and generalize to new domains [1–4]. I would request the authors to point out if I misunderstood their novel contributions.
2. The authors did not show their performance on any real-world data or compared with any existing baselines.
3. The authors mentioned generative augmentation models, but the process of generating augmented data is not properly discussed.
4. The paper "Do causal predictors generalize better to new domains?" [5] argues that causal parents are not always good predictors and demonstrates better performance using non-causal variables as input features. I suggest that the authors evaluate their algorithm on those datasets.


References:

[1] Peters, J., Bühlmann, P., & Meinshausen, N. (2016). Causal inference by using invariant prediction: identification and confidence intervals. J. R. Stat. Soc. B, 78(5), 947–1012.

[2] Arjovsky, M., Bottou, L., Gulrajani, I., & Lopez-Paz, D. (2019). Invariant Risk Minimization. arXiv preprint arXiv:1907.02893.

[3] Magliacane, S., van Ommen, T., Claassen, T., Schölkopf, B., & Mooij, J. (2018). Domain adaptation by using causal inference to predict invariant conditional distributions. Adv. Neural Inf. Process. Syst. (NeurIPS) 31.

[4] Subbaswamy, A., Schulam, P., & Saria, S. (2019). Preventing failures due to dataset shift: Learning predictive models that transport. Int. Conf. Artif. Intell. Stat. (AISTATS), PMLR.

[5] Nastl, V., & Hardt, M. (2024). Do causal predictors generalize better to new domains? Adv. Neural Inf. Process. Syst. (NeurIPS) 37, 31202–31315.

## Minor:
1. The authors mention a stochastic optimization problem in the introduction, but it remains unclear until later in the paper.
2. Line 368: Figures 3,4 appear to be mislabeled incorrectly as Figure 7,8.
3. The connection between distributionally robust optimization (DRO) and interventionally robust optimization is unclear. DRO is a well-established concept; the authors should clarify whether they refer to the same idea and formally establish the connection.
5. Line 149: the function $h$ outputs $\mathbb{R}^{|T|}$. The function $Q$ takes as input $\mathbb{R}^{|T|} \times \mathbb{R}^{|V \setminus T|}$. If the output of $h$ is used as input to $Q$, will that be consistent?
6. What does id mean in Equation (2)? It is also unclear what $i, k$ refer to in $\hat{T}_i^k$.
7. The authors should provide a workflow diagram illustrating their architecture.

**Questions:**

Below I share my questions:

## Questions:
1. What is $ \bar{R} $ in line 153? Is it defined?
3. What is $ \mu $ in Equation (1), and why is the expectation taken with respect to it?
4. Can the optimization problem not be solved with a different model and a different loss function? Is it continuous only under the G-causal Wasserstein distance with a normalizing flow?

---

### Note · Authors · 2025-11-20

**Comment:**

Dear reviewers,

Thank you for taking the time to evaluate our submission and for providing such valuable feedback. Your remarks motivate us to find a better way of framing our contribution within the relevant literature and to strengthen the practical relevance of our findings. Due to the time constraints of the review process, we are currently unable to address all the points you raised in a satisfactory way. We have therefore decided to withdraw the paper and to take the necessary time to implement all the necessary changes. Thank you again for your time.

**Withdrawal Confirmation:**

I have read and agree with the venue's withdrawal policy on behalf of myself and my co-authors.